# Epidemiology of Primary Epithelial Salivary Gland Tumors in Southern Poland—A 26-Year, Clinicopathologic, Retrospective Analysis

**DOI:** 10.3390/jcm10081663

**Published:** 2021-04-13

**Authors:** Michał Gontarz, Jakub Bargiel, Krzysztof Gąsiorowski, Tomasz Marecik, Paweł Szczurowski, Jan Zapała, Grażyna Wyszyńska-Pawelec

**Affiliations:** Department of Cranio-Maxillofacial Surgery, Jagiellonian University Medical College, 31-008 Cracow, Poland; jakub.bargiel@uj.edu.pl (J.B.); krzysztof.gasiorowski@uj.edu.pl (K.G.); tomasz.marecik@uj.edu.pl (T.M.); pawel.szczurowski@uj.edu.pl (P.S.); jan.zapala@uj.edu.pl (J.Z.); grazyna.wyszynska-pawelec@uj.edu.pl (G.W.-P.)

**Keywords:** salivary gland neoplasms, salivary glands, epidemiology, incidence, pleomorphic adenoma, Warthin tumor, adenolymphoma, mucoepidermoid carcinoma, adenoid cystic carcinoma

## Abstract

(1) Background: Epidemiological studies of epithelial salivary gland neoplasms are difficult to conduct effectively due to tumor rarity, histological heterogeneity, tumor location diversity, and a lack of national registries collecting data. This study presents 26 years of epidemiological data from a single institution in southern Poland that estimates incidence rates of primary epithelial salivary gland tumors. (2) Methods: The charts of 805 patients with epithelial salivary gland tumors were retrospectively reviewed. (3) Results: Pleomorphic adenomas occurred less frequently in elderly patients; however, Warthin tumors were more common (*p* < 0.001). Pediatric patients mainly suffered from mucoepidermoid carcinoma. The estimated crude and European age-standardized incidence rates of all primary epithelial salivary gland tumors were 6.7 and 6.02 per 100,000 population, respectively. The incidence rates of salivary gland tumors increased in recent years; however, this is attributed to an increase in benign tumors (*p* < 0.001). (4) Conclusions: The incidence of primary epithelial salivary gland tumors in southern Poland has increased over the past 26 years. This increase is attributed to a rise in the number of patients with benign tumors, particularly Warthin tumors in elderly patients. Moreover, the incidence of malignant salivary gland tumors appears to be higher in pediatric patients.

## 1. Introduction

Salivary gland tumors are rare neoplasms that comprise approximately 3–6% of all head and neck tumors. The incidence rate varies from 0.4 to 13.5 cases per 100,000 population, with most salivary gland tumors being benign [1]. Generally, most salivary gland tumors occur in the parotid gland, followed by the submandibular and minor salivary glands. However, in some cases, salivary gland tumors are also observed to originate from the upper aerodigestive tract, such as the lacrimal gland or cervical lymph nodes [2,3]. The current 2017 World Health Organization (WHO) salivary gland tumor classification guidelines distinguish 31 different primary neoplasms [4]. The most common histological types are pleomorphic adenoma (PA), Warthin’s tumor (WT), mucoepidermoid carcinoma (MEC), and adenoid cystic carcinoma (ACC) [5,6,7]. However, large histological heterogeneity with a wide range of tumor locations may cause diagnostic and therapeutic problems in some cases.

While several large series of salivary gland tumors have been published in the English literature, only a few are from European institutions [2,5,6,7,8,9,10,11]. The majority of the epidemiological studies available assess the incidence rate of malignant salivary gland neoplasms [12,13]. Moreover, data from the Polish National Cancer Registry are limited to malignant major salivary gland tumors (ICD-10: C07 malignant neoplasm of the parotid gland and C08 malignant neoplasm of other and unspecified major salivary glands), thus only representing a collection of epithelial tumors as well as nonepithelial, secondary, and hematolymphoid malignant neoplasms. Available data from malignant minor salivary gland tumors are limited. Some papers do estimate an incidence rate of benign salivary gland tumors [14,15]. However, those studies also include nonepithelial tumors and are limited only to parotid neoplasms. This is the first report in the English literature on a large series of primary epithelial salivary gland tumors in Poland. This paper presents epidemiological data from a single institution over the last 26 years and estimates the incidence rates of malignant and benign tumors in southern Poland.

## 2. Materials and Methods

Prior to data collection and analysis, Institutional Review Board approval was obtained (No: 122.6120.287.2016). Because the study was confined to a chart review and retrospective in nature, the Institutional Review Board waived the need for patient consent so long as all personal information remained confidential.

A retrospective chart review of all patients treated for salivary gland tumors at our Department between January 1994 and December 2019 was conducted. Patients meeting any of the following exclusion criteria were excluded from the study: tumor-like lesions (chronic inflammations, cysts, sclerosing polycystic adenosis, benign lymphoepithelial lesions, IgG4-related disease, Sjögren syndrome, sarcoidosis, tuberculosis, oncocytic metaplasia, toxoplasmosis, cat scratch disease, Masson’s tumor, mononucleosis), secondary tumors (malignant melanoma, skin squamous cell carcinoma, follicular thyroid cancer, clear cell renal cell carcinoma, undifferentiated lung carcinoma), nonepithelial tumors (lipoma, hemangioma, Schwannoma), and lymphomas. The remaining patients with primary epithelial salivary gland neoplasms were included in the study group. Study group patient charts were retrospectively reviewed and evaluated based on demographic characteristics and histopathological aspects.

According to the Polish National Cancer Registry (PNCR) database, 556 cases with the ICD-10 codes C07 and C08 were diagnosed between 2009 and 2017 in the province of Małopolska in southern Poland. The crude incidence rate (CR) was 0.89 per 100,000 population and the European age-standardized incidence rate (EASR) was 0.8 based on the European Standard Population (ESP-2013). The PNCR incidence rates for Małopolska were scaled by the ratio of C07 and C08 neoplasms to all neoplasms (malignant and benign, minor and major salivary gland), based on the data collected by the authors to estimate incidence rates for different groups of patients. The Poisson distribution was used to compare the incidence rates from different time periods in the study.

The study cohort was divided into groups based on sex (female or male), age (0–19, 20–59, and over 60 years old), and time period (1994–2000, 2001–2010, and 2011–2019) to estimate any differences between them. Comparison of the qualitative variables for each patient group was performed using the χ^2^ test (with the Yates’s correction applied for 2 × 2 tables) or the Fisher’s exact test for small values. The Mann–Whitney *U* test was conducted to compare quantitative variables among two groups, while the Kruskal–Wallis test was conducted for three or more groups. If statistically significant differences were detected after the Kruskal–Wallis test was completed, post-hoc analysis using the Dunn’s test was performed to further identify the statistically significant groups. A significance level of 0.05 was used for all tests (α = 0.05). All statistical analyses were performed using R v4.0.2 (R Core Team (2020); R: A language and environment for statistical computing. R Foundation for Statistical Computing, Vienna, Austria).

## 3. Results

A total of 940 patients were identified as having received treatment for salivary gland tumors within the study date range. Of these 940 patients, 72 patients were excluded for tumor-like lesions, 30 patients were excluded for secondary tumors, 20 patients were excluded for nonepithelial tumors, and 13 patients were excluded for lymphomas. Therefore, the resulting study group totaled 805 cases of primary epithelial salivary gland neoplasms. All patients in the study population were of Caucasian ethnicity.

### 3.1. Histological Type and Distribution

During the 26-year period reviewed, 805 patients underwent surgical treatment for primary epithelial salivary gland tumors. Among these patients, 566 (70.3%) had benign tumors and 239 (29.7%) had malignant tumors. The ratio of benign to malignant neoplasm was 2.37:1. PA was the most common tumor and comprised 66.4% (376/566) of all benign tumors and 46.7% (376/805) of all salivary neoplasms. The second most common was WT, accounting for 24% (136/566) of all benign tumors and 16.9% (136/805) of all tumors in the study group. Of the 239 malignancies, ACC (73/239, 30.5%) was the most common histological type, followed by MEC (58/239, 24.3%). The details of histological type and sites of tumors are shown in Table 1 and Table 2.

### 3.2. Age

The age of patients ranged from 8 to 90 years, with a mean age of 53 years and a median age of 55. Table 3 shows the relationship between the clinicopathological parameters among pediatric/adolescent, adult, and elderly patients.

In elderly patients, PA occurred less frequently, but WT was more common (*p* < 0.001). The percentage of PAs decreased with each patient age group, whereas the percentage of WTs increased. Pediatric patients primarily suffered from MEC, and the percentage of those tumors decreased with age group. In contrast, the incidence of ACC increased with each age group (*p* = 0.002). The results also indicated a higher incidence of malignant neoplasms among the youngest patients (*p* = 0.051).

### 3.3. Sex

Of the 805 patients, 441 (54.8%) were female, and 364 (45.2%) were male. The female-to-male ratio was 1.2:1. A comparison of patient characteristics between female and male patients is presented in Table 4.

In the study group, males were older than females (*p* = 0.047), WT was observed more often in males, and PA was observed more often in females (*p* < 0.001). ACC accounted for a larger percentage of malignant tumors in females than males; however, males often suffered from types of carcinomas other than ACC and MEC (*p* = 0.028).

### 3.4. Time Period

This cohort was divided into three time periods: 1994–2000, 2001–2010, and 2011–2019. Changes of clinicopathological factors in each time period are presented in Table 5.

With each subsequent time period, the age of operative patients increased (*p* < 0.001). Likewise, the number of parotid gland tumors increased over the years (*p* < 0.001). In the most recent time period, a higher incidence of benign neoplasms was observed (*p* = 0.003). This is mainly attributed to an increase in the number of patients operated on for WT (*p* < 0.001). The proportion of PA among benign tumors was the highest in the 1994–2000 period and decreased over time (*p* = 0.004). The highest number of ACC cases among malignant neoplasms was observed between 2001 and 2010, while the number of patients treated for MEC increased over the last ten years (*p* = 0.05).

### 3.5. Incidence

The estimated Małopolska province incidence rates, CR and EASR, were 6.7 and 6.02 per 100,000, respectively (Table 6). For benign tumors, the CR was 4.71, and the EASR was 4.23. The CR and EASR for malignant tumors were 1.99 and 1.79, respectively. Primary epithelial salivary gland tumors in children and adolescents were very rare with a CR and EASR of only 0.49 for both. The CR and EASR increased with the age of patients and were the highest in elderly patients at 19.33 and 18.8, respectively. The CR and EASR for benign and malignant tumors were similar between males and females. The incidence rates of salivary gland tumors increased in recent years; however, this was due to an increase in the number of benign tumors (*p* < 0.001). The incidence of malignant neoplasms remained at the same level over all three time periods (*p* = 0.867) (Table 7).

## 4. Discussion

Epidemiological studies of epithelial salivary gland neoplasms are difficult to conduct effectively due to tumor rarity, histological heterogeneity, tumor location diversity, and a lack of national registries collecting data, especially for benign tumors. Since 2014, the Polish National Major Salivary Gland Benign Tumors Registry has been collecting data on benign major salivary gland tumors in Poland [16]. However, the registry also contains data on nonepithelial tumors and does not include the incidence of benign minor salivary gland neoplasms, which amounted to 82 cases constituting 14.5% of benign and 10.2% of all epithelial neoplasms. The majority of available epidemiological studies are single-institution reviews from around the world. Although some papers include nonepithelial neoplasms, many studies only present a descriptive epidemiology of salivary gland tumors without a statistical assessment [2,6,7,8,9,10,11,15].

In the vast majority of cases, epithelial salivary gland tumors are benign, which accounted for 70.3% of neoplasms in our study. This is comparable to results from other centers in the world, where benign neoplasms comprised a range from 57.7% to 77.7% of cases [2,5,7,8,10,11,17]. Only Tilakaratne et al. observed a higher percentage of malignant tumors (50.1%) in the Sri Lankan population [9]. In all present epidemiological studies, PA is the most common salivary gland tumor [2,5,7,8,9,10,11,17]. Likewise, PAs accounted for 66.4% of benign and 46.7% of all neoplasms in our analysis. In 75.3% (283/376) of cases, PAs were located in the parotid gland, followed by 14.6% (55/376) in the palate. PAs were the only histologically benign neoplasms observed in our group of pediatric patients. The percentage of PAs in the benign salivary gland tumors group decreased with each subsequently older patient age group (*p* < 0.001). There was also a significantly more frequent occurrence of PA among women (*p* < 0.001), which is similar to other studies [5,6,9]. The number of patients receiving surgical intervention due to PA has increased in recent years. However, the percentage of PAs relative to other benign neoplasms has decreased compared to the end of the 20th century (*p* = 0.004).

In accordance with the literature, WT was the second most common primary epithelial salivary gland tumor in our study [2,5,6,10,17]. In our results, WT constituted 24% of all benign tumors (136/566) and 16.9% of tumors in the entire study group (136/805). WT was more common in patients over the age of 60, especially among men (*p* < 0.001). This was in line with other studies [2,5,6,8,11]. Recently, we observed an increase in the number of patients receiving surgical intervention due to WT when compared to other benign salivary gland tumors (*p* < 0.001). This increase in the incidence of WT may not only be the result of smoking; this increase may also be correlated to obesity, an increasingly major problem in modern society. In the Małopolska province, 16% of the adult population suffers from obesity [18]. The incidence of obesity increases with population age, the same as the risk for WT. This can be corroborated by Kadletz et al., where they observed a statistically higher BMI in patients with WT when compared to patients treated for other benign salivary gland tumors [19]. Additionally, higher incidence of WT can be explained by higher sensitivity in their detection in imaging examinations. WT is known to have high 18-fluorodeoxyglucose (FDG) uptake and therefore might be found in the parotid gland as an incidentaloma in PET/CT [20]. Schwalje et al. proved that the growth rate of WT is slow, especially in patients over 75 years old [21]. For this reason, conservative management (wait and see policy) of WT, diagnosed with fine-needle aspiration cytology (FNAC), is an alternative to surgery, especially for patients suffering from cancers treated with nonsurgical modality. In the current study, in the vast majority of cases, WT was located in the parotid gland. In five cases, WT was found in the cervical lymph nodes; in four of these patients, the tumors were synchronous with oral squamous cell carcinoma, suggesting neck metastases. Extraparotid localization is uncommon and only observed in up to 8% of all WTs [3]. In other epidemiological studies, the authors did not describe WT in the cervical lymph nodes but did present single cases of WT in the submandibular and minor salivary glands [2,5,7,8,9]. However, none of the epidemiological studies found WT in the sublingual gland. Our group did not observe WT in the remaining major and minor salivary glands.

According to the WHO and most epidemiological studies, the most common malignant salivary gland tumor is MEC [1,5,6,7,8,9,11,13,22]. However, in some studies, particularly from European centers and our study results, the most common malignant tumor was ACC [2,10,12,17,23]. In our study, ACC accounted for 30.5% (73/239) of malignancies and 9% (73/805) of all tumors. ACC was mainly located in the minor salivary glands (73.6%, 53/73), most often in the maxilla and palate (73.6%; 39/53). There were no cases of ACC in the pediatric and adolescent group. Only single cases of ACC in pediatric patients have been described in the literature [9,24,25]. However, we did observe that the incidence of ACC increased with each subsequent patient age group (*p* = 0.002), and peaked in the adult patient group (20–59 yo). This neoplasm was also observed more often in females (*p* = 0.028), the same as in other studies [12,19,23]. The highest percentage of patients with ACC was observed from 2001–2010. Presently, the number of patients treated for ACC has decreased (*p* = 0.05). Additionally, Sentani et al. observed a slight decrease in the number of patients with ACC in the last two decades [26]. This is likely due to the current more accurate histological differential diagnosis.

The second most common malignant tumor was MEC, constituting 24.3% (58/239) of malignant neoplasms and 7.2% (58/805) of all neoplasms. Similar to ACCs, 70.7% (41/58) of MECs were usually located in the minor salivary glands, most often in the maxilla and palate (56.1%; 23/41). MECs were the most common malignant neoplasm in the pediatric and adolescent group, and their incidence decreased with each older patient age group (*p* = 0.002). Fortunately, the majority of MECs in pediatric patients are low-grade with favorable outcomes [24,25,27]. Nevertheless, an increase in the incidence of MEC had been observed (*p* = 0.05) in recent years (2011–2019).

The majority of primary epithelial salivary gland tumors are located in the major salivary glands, most frequently in the parotid gland [2,5,6,7,8,10,17]. Likewise, in our study, tumors of the major salivary glands occurred in 586 (72.8%) cases, 81.7% (479/586) of which were parotid tumors. An increase in the number of patients recently operated on for major salivary gland neoplasms was observed in this study (*p* < 0.001). Certain anatomical locations are predisposed to benign and malignant salivary gland neoplasms occurring [11,28]. Most parotid gland tumors are benign, ranging from 68.7% to 88% of tumors [2,5,6,7,8,9,10]. This is concurrent with our results, where 77.6% (455/586) of tumors were benign. Benign neoplasms are reportedly more common in the submandibular gland [2,5,6,7,8,10]. However, in both our analysis and in Tilakaratne et al., an almost equal percentage of malignant and benign tumors was found in the submandibular gland [9]. Tumors of the sublingual gland are very rare, accounting for only about 1% of all salivary gland tumors [2,6,7,9,10]. In contrast to the parotid and submandibular glands, the percentage of malignant sublingual gland tumors is much higher, ranging from 75% to 100% in the literature [2,5,6,7,9,10]. In our study, neoplasms of the sublingual gland accounted for approximately 0.9% (7/805) of cases, of which 71.4% (5/7) were malignant. In accordance with the literature, the most common malignant neoplasm in the sublingual gland was an ACC [2,5,6,7,9]. On the other hand, the only histologically benign tumor in the sublingual gland was a PA, the same as in other studies [2,6,7,9].

Minor salivary gland tumors accounted for a relatively high percentage of cases 26.6% (214/805) in our study. In various studies in the literature, the incidence of minor salivary gland tumors ranges from 14% to 28% of cases [2,5,6,8,10,26]. In our analysis, most minor salivary gland neoplasms were malignant (61.7%, 132/214). Our results correspond to studies from other clinical centers [2,5,10]. A lower percentage of malignant neoplasms is observed in studies from pathology centers [7,9]. This is likely due to the fact that many benign minor salivary gland tumors are treated in dental surgery outpatient clinics, and therefore their epidemiology is underestimated [28]. In palatal and buccal mucosa tumors, a similar percentage of benign and malignant neoplasms were observed. The malignancy of minor salivary gland tumors increased for lower locations within the oral cavity. In this cohort, 100% of tumors located in the lower lip, tongue, floor of the mouth, or gingiva were malignant. Likewise, 88.9% (8/9) of tumors in the retromolar triangle were also malignant. Similar observations were found in other studies [11,29].

Recently, the CR and EASR of salivary gland tumors have increased; however, this can be largely attributed to the increase in the number of patients treated for benign tumors. Over the past 26 years in the Małopolska province, the number of patients treated for malignant salivary gland tumors increased; however, the CR and EASR remained stable (*p* = 0.867). This was probably caused by population growth and aging. The incidence of malignant salivary gland tumors also remained stable in other countries, such as the USA, the Netherlands, Denmark, China, and Japan [12,13,19,23,26]. There are a few studies in the literature assessing the incidence rate of benign salivary gland neoplasms, but only in the major glands [14,15,30]. In the Pinkston and Cole study, the age-standardized incidence rate for benign major salivary gland tumors was 4.72, which corresponds to the results of our study, where the EASR for all benign salivary gland tumors was 4.23 [30]. Epidemiological studies of primary epithelial salivary gland neoplasms exhibit a large degree of bias, which is often due to the fact that nonepithelial, secondary, and hematolymphoid tumors are not excluded from the analysis. On the other hand, single-center studies are not of sufficient statistical power to estimate the number of new cases in the present population. In addition, studies originating from oncological centers will be biased by a greater percentage of patients being treated for malignant tumors. Moreover, data from the National Cancer Registers only concern malignant tumors and often only include data on secondary neoplasms [22]. Primary benign minor salivary gland tumor incidence is often underestimated by the frequent treatment of these tumors in small outpatient dental surgery centers. The most accurate epidemiological data would likely be obtained from the national registries that are collecting data from pathology centers because they consider whether the tumor is primary, secondary, or recurrent.

## 5. Conclusions

In conclusion, this study found an increase in the incidence of primary epithelial salivary gland tumors in southern Poland. This increase was mainly attributed to the increase in the number of new cases of benign tumors, especially WT in elderly patients. Moreover, the incidence of malignant salivary gland tumors appears to be higher in pediatric patients than in adult and elderly patients. ACC was the most common malignant tumor, although the incidence of MEC has increased recently.

## Figures and Tables

**Table 1 jcm-10-01663-t001:** Location and histological type of benign salivary gland tumors.

	Major Glands No. (%)			Minor Glands No. (%)				Cervical Lymph Nodes	Total No. (%)
	Parotid	Submandibular	Sublingual	Palate	Buccal	Lips	Retromolar		
Pleomorphic adenoma	283	20	2	55	13	2	1		376 (66.4)
Warthin tumor	131							5	136 (24)
Basal cell adenoma	18			1	3				22 (3.9)
Myoepithelioma	12	1		3	2				18 (3.2)
Oncocytoma	8	1							9 (1.6)
Canalicular adenoma	2			1	1				4 (0.7)
Tubular adenoma	1								1 (0.2)
Total No. (%)	455 (80.4)	22 (3.8)	2 (0.4)	60 (10.6)	19 (3.3)	2 (0.4)	1 (0.2)	5 (0.9)	566 (100)

**Table 2 jcm-10-01663-t002:** Location and histological type of malignant salivary gland tumors.

	Major Salivary Glands			Minor Salivary Glands								Total No. (%)
	Parotid	Submandibular	Sublingual	Palate	Buccal	Maxilla	Lips	Retromolar	FOM	Tongue	Lower Gingiva	
Adenoid cystic carcinoma	11	6	3	17	5	22	4	1		4		73 (30.5)
Mucoepidermoid carcinoma	13	4		19	4	4	1	6	3	2	2	58 (24.3)
Adenocarcinoma, NOS	10	4		3	5	2	3		1			28 (11.7)
Ca ex PA	16	1	1	3	2	1		1				25 (10.5)
Acinic cell carcinoma	6	1		2	5							14 (5.9)
Squamous cell carcinoma	5	7	1									13 (5.5)
Polymorphous adenocarcinoma	2			4	1	2						9 (3.7)
Salivary duct carcinoma	3											3 (1.3)
Myoepithelial carcinoma	3											3 (1.3)
Undifferentiated carcinoma	2	1										3 (1.3)
Oncocytic carcinoma	2											2 (0.8)
Epithelial-myoepithelial carcinoma	1											1 (0.4)
Lymphoepithelial carcinoma	1											1 (0.4)
Cystadenocarcinoma				1								1 (0.4)
Large cell carcinoma									1			1 (0.4)
Small cell carcinoma	1											1 (0.4)
Basal cell adenocarcinoma	1											1 (0.4)
Mammary analogue secretory carcinoma	1											1 (0.4)
Mucinous adenocarcinoma									1			1 (0.4)
Total No. (%)	78 (32.6)	24 (10.0)	5 (2.1)	49 (20.5)	22 (9.2)	31 (13)	8 (3.4)	8 (3.4)	6 (2.5)	6 (2.5)	2 (0.8)	239 (100)

FOM—floor of the mouth; NOS—not otherwise specified; Ca ex PA—carcinoma ex pleomorphic adenoma.

**Table 3 jcm-10-01663-t003:** Clinicopathological characteristics by age.

Parameter		Age			*p*-Value
		0–19 yo	20–59 yo	>60 yo	
**Sex**	Male	7 (33.33%)	199 (42.98%)	158 (49.22%)	*p* = 0.122
	Female	14 (66.67%)	264 (57.02%)	163 (50.78%)	
**Site**	Major salivary gland	12 (57.14%)	341 (73.65%)	232 (72.27%)	*p* = 0.382
	Minor salivary gland	9 (42.86%)	120 (25.92%)	86 (26.79%)	
	Cervical lymph nodes	0 (0.00%)	2 (0.43%)	3 (0.93%)	
**Minor salivary gland**	Palate	6 (66.67%)	63 (52.50%)	40 (46.51%)	*p* = 0.674
	Buccal	2 (22.22%)	22 (18.33%)	17 (19.77%)	
	Others	1 (11.11%)	35 (29.17%)	29 (33.72%)	
**Major salivary gland**	Parotid	10 (83.33%)	309 (90.62%)	214 (92.24%)	*p* = 0.383
	Others	2 (16.67%)	32 (9.38%)	18 (7.76%)	
**Malignancy**	Benign	11 (52.38%)	338 (73.00%)	217 (67.60%)	*p* = 0.051
	Malignant	10 (47.62%)	125 (27.00%)	104 (32.40%)	
**Histological type**	Pleomorphic adenoma	11 (52.38%)	269 (58.10%)	96 (29.91%)	***p* < 0.001**
	Warthin’s tumor	0 (0.00%)	38 (8.21%)	98 (30.53%)	
	Mucoepidermoid carcinoma	7 (33.33%)	34 (7.34%)	17 (5.30%)	
	Adenoid cystic carcinoma	0 (0.00%)	41 (8.86%)	32 (9.97%)	
	Others	3 (14.29%)	81 (17.49%)	78 (24.30%)	
**Benign tumors**	Pleomorphic adenoma	11 (100.00%)	269 (79.59%)	96 (44.24%)	***p* < 0.001**
	Warthin’s tumor	0 (0.00%)	38 (11.24%)	98 (45.16%)	
	Others	0 (0.00%)	31 (9.17%)	23 (10.60%)	
**Malignant tumors**	Adenoid cystic carcinoma	0 (0.00%)	41 (32.80%)	32 (30.77%)	***p* = 0.002**
	Mucoepidermoid carcinoma	7 (70.00%)	34 (27.20%)	17 (16.35%)	
	Others	3 (30.00%)	50 (40.00%)	55 (52.88%)	

yo—years old.

**Table 4 jcm-10-01663-t004:** Clinicopathological characteristics by sex.

Parameter		Sex		*p*-value
		Male	Female	
**Age**	mean±SD	54.38 ± 15.86	51.88 ± 16.84	***p* = 0.047**
	median	57	54	
	quartiles	44–66	40–65	
**Time period**	1994–2000	75 (20.60%)	87 (19.73%)	*p* = 0.184
	2001–2010	99 (27.20%)	146 (33.11%)	
	2011–2019	190 (52.20%)	208 (47.17%)	
**Site**	Major salivary gland	270 (74.18%)	315 (71.43%)	*p* = 0.492
	Minor salivary gland	91 (25.00%)	124 (28.12%)	
	Cervical lymph nodes	3 (0.82%)	2 (0.45%)	
**Minor salivary gland**	Palate	44 (48.35%)	65 (52.42%)	*p* = 0.642
	Buccal	20 (21.98%)	21 (16.94%)	
	Others	27 (29.67%)	38 (30.65%)	
**Major salivary gland**	Parotid	247 (91.48%)	286 (90.79%)	*p* = 0.884
	Others	23 (8.52%)	29 (9.21%)	
**Malignancy**	Benign	262 (71.98%)	304 (68.93%)	*p* = 0.388
	Malignant	102 (28.02%)	137 (31.07%)	
**Histological type**	Pleomorphic adenoma	144 (39.56%)	232 (52.61%)	***p* < 0.001**
	Warthin’s tumor	92 (25.27%)	44 (9.98%)	
	Mucoepidermoid carcinoma	26 (7.14%)	32 (7.26%)	
	Adenoid cystic carcinoma	22 (6.04%)	51 (11.56%)	
	Others	80 (21.98%)	82 (18.59%)	
**Benign tumors**	Pleomorphic adenoma	144 (54.96%)	232 (76.32%)	***p* < 0.001**
	Warthin’s tumor	92 (35.11%)	44 (14.47%)	
	Others	26 (9.92%)	28 (9.21%)	
**Malignant tumors**	Adenoid cystic carcinoma	22 (21.57%)	51 (37.23%)	***p* = 0.028**
	Mucoepidermoid carcinoma	26 (25.49%)	32 (23.36%)	
	Others	54 (52.94%)	54 (39.42%)	

**Table 5 jcm-10-01663-t005:** Clinicopathological characteristics by time period.

Parameter		Time Period			*p*-Value
		1994–2000 (A)	2001–2010 (B)	2011–2019 (C)	
**Age**	Mean ± SD	48.35 ± 15.06	52.5 ± 16.66	55.23 ± 16.46	***p* < 0.001**
	median	49	55	58	
	quartiles	39–60	42–65	44–67	C > B > A
**Sex**	Male	75 (46.30%)	99 (40.41%)	190 (47.74%)	*p* = 0.184
	Female	87 (53.70%)	146 (59.59%)	208 (52.26%)	
**Site**	Major salivary gland	116 (71.60%)	183 (74.69%)	286 (71.86%)	*p* = 0.524
	Minor salivary gland	46 (28.40%)	59 (24.08%)	110 (27.64%)	
	Cervical lymph nodes	0 (0.00%)	3 (1.22%)	2 (0.50%)	
**Minor salivary gland**	Palate	22 (47.83%)	26 (44.07%)	61 (55.45%)	*p* = 0.416
	Buccal	11 (23.91%)	10 (16.95%)	20 (18.18%)	
	Others	13 (28.26%)	23 (38.98%)	29 (26.36%)	
**Major salivary gland**	Parotid	95 (81.90%)	168 (91.80%)	270 (94.41%)	***p* < 0.001**
	Others	21 (18.10%)	15 (8.20%)	16 (5.59%)	
**Malignancy**	Benign	98 (60.49%)	170 (69.39%)	298 (74.87%)	***p* = 0.003**
	Malignant	64 (39.51%)	75 (30.61%)	100 (25.13%)	
**Histological type**	Pleomorphic adenoma	78 (48.15%)	105 (42.86%)	193 (48.49%)	***p* < 0.001**
	Warthin’s tumor	10 (6.17%)	44 (17.96%)	82 (20.60%)	
	Mucoepidermoid carcinoma	16 (9.88%)	11 (4.49%)	31 (7.79%)	
	Adenoid cystic carcinoma	19 (11.73%)	31 (12.65%)	23 (5.78%)	
	Others	39 (24.07%)	54 (22.04%)	69 (17.34%)	
**Benign tumors**	Pleomorphic adenoma	78 (79.59%)	105 (61.76%)	193 (64.77%)	***p* = 0.004**
	Warthin’s tumor	10 (10.20%)	44 (25.88%)	82 (27.52%)	
	Others	10 (10.20%)	21 (12.35%)	23 (7.72%)	
**Malignant tumors**	Adenoid cystic carcinoma	19 (29.69%)	31 (41.33%)	23 (23.00%)	***p* = 0.05**
	Mucoepidermoid carcinoma	16 (25.00%)	11 (14.67%)	31 (31.00%)	
	Others	29 (45.31%)	33 (44.00%)	46 (46.00%)	

**Table 6 jcm-10-01663-t006:** Crude and European age-standardized incidence rates (per 100,000) for salivary gland tumors in southern Poland.

		PNCR Data			Study Data									
					Benign and Malignant Tumors				Benign Tumors			Malignant Tumors		
		C07, C08	CR	EASR	C07, C08	Total No	CR	EASR	Total No	CR	EASR	Total No	CR	EASR
**All cases**		556	0.89	0.8	107	805	6.7	6.02	566	4.71	4.23	239	1.99	1.79
	1994–2000	72	1.12	1.17	32	162	5.67	5.92	98	3.43	3.58	64	2.24	2.34
**Time**	2001–2010	293	0.90	0.83	34	245	6.49	5.98	170	4.5	4.15	75	1.99	1.83
	2011–2019	191	0.81	0.70	41	398	7.86	6.8	298	5.89	5.09	100	1.98	1.71
	0–19	11	0.07	0.07	3	21	0.49	0.49	11	0.26	0.26	10	0.23	0.23
**Age**	20–59	200	0.56	0.58	55	463	4.71	4.88	338	3.44	3.56	125	1.27	1.32
	>60	345	2.95	2.87	49	321	19.33	18.8	217	13.06	12.71	104	6.26	6.09
**Sex**	Female	278	0.87	0.73	56	441	6.85	5.75	304	4.72	3.96	137	2.13	1.79
	Male	278	0.92	0.93	51	364	6.57	6.64	262	4.73	4.78	102	1.84	1.86

PNCR—Polish National Cancer Registry; CR—crude rate; EASR—European age-standardized rate.

**Table 7 jcm-10-01663-t007:** Crude and European age-standardized incidence rates (per 100,000) for salivary gland tumors characteristics by time period in southern Poland.

	CR–Crude Rate						EASR–European Age-Standardized Rate					
	1994–2000	2001–2010	2011–2019	1994–2000 vs. 2001–2010	1994–2000 vs. 2011–2019	2001–2010 vs. 2011–2019	1994–2000	2001–2010	2011–2019	1994–2000 vs. 2001–2010	1994–2000 vs. 2011–2019	2001–2010 vs. 2011–2019
**Benign and malignant tumor**	5.67	6.49	7.86	***p* = 0.01**	***p* < 0.001**	***p* < 0.001**	5.92	5.98	6.8	*p* = 0.435	***p* = 0.009**	***p* < 0.001**
**Benign tumor**	3.43	4.5	5.89	***p* < 0.001**	***p* < 0.001**	***p* < 0.001**	3.58	4.15	5.09	***p* = 0.023**	***p* < 0.001**	***p* < 0.001**
**Malignant tumor**	2.24	1.99	1.98	*p* = 0.907	*p* = 0.908	*p* = 0.532	2.34	1.83	1.71	*p* = 0.997	*p* = 1	*p* = 0.867

## Data Availability

Restrictions apply to the availability of these data. Data was obtained from patients treated at the Department of Cranio-Maxillofacial Surgery, Cracow, Poland, and cannot be shared, in accordance with the General Data Protection Regulation (EU) 2016/679.

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
