# Peer review of "Epidemiology of Primary Epithelial Salivary Gland Tumors in Southern Poland—A 26-Year, Clinicopathologic, Retrospective Analysis"

_jcm, 2021, doi:10.3390/jcm10081663_

Round 1

Reviewer 1 Report

The manuscript describes the epidemiology of primary salivary gland tumors in southern Poland over a 26 year period.

The study is descriptive and lacks any hypothesis or question to be answered. As such the study lacks any novelty. The paper would better suited to a local otolaryngology journal.

Author Response

Reviewer 1

The manuscript describes the epidemiology of primary salivary gland tumors in southern Poland over a 26 year period.

The study is descriptive and lacks any hypothesis or question to be answered. As such the study lacks any novelty. The paper would better suited to a local otolaryngology journal.

Author comments

Thank you for your assessment. However, we believe that the epidemiology of primary salivary gland tumors from Poland in English literature is missing and information from this paper are likely to be of great interest to the researchers from all of the World. Most epidemiological studies of salivary gland tumors does not concern only primary epithelial tumors, but usually include non-epithelial, hematolymphoid as well as secondary tumors. Majority of papers present a descriptive epidemiology of salivary gland tumors without a statistical assessment. In this study epidemiological data are represented on a selected group of patients treated for primary epithelial salivary gland tumors. Also all patients in the study population were of Caucasian ethnicity. In addition, the paper describes changes in the incidence rate of such tumors as Warthin tumor or adenoid cystic carcinoma which could be compared to those observed in other countries.

Reviewer 2 Report

General Comments: This manuscript describes the characteristics and incidence of salivary gland tumors over a 26 year period in Malopolska province in Poland. Overall this manuscript provides a strong resource dataset; however, there are some clarifications necessary to fully comprehend the data analysis.

Major concerns:

  1. Abstract: There is a disconnect between the results and conclusion text in that there are specific conclusions stated when the data are not described in the results
  2. Table 3 needs a legend to explain what the p value is comparing. As presented, the p value appears to denote differences within a category (e.g. histological type). However, the results text seems to describe comparisons of one aspect of a category “In elderly patients, PA occurred less frequently, but WT was more common (p < 0.001)”. The p value is listed in the same row as PA, which makes this sentence very confusing. A similar issue exists for Tables 4 and 5.
  3. Table 4 is exactly the same as Table 3 and does not report the characteristics by sex
  4. Is there information on recurrence or 5 year follow-up?
  5. Discussion: obesity link with WT should include obesity rates in Malopolska province.

Author Response

Reviewer 2

Comments and Suggestions for Authors

General Comments: This manuscript describes the characteristics and incidence of salivary gland tumors over a 26 year period in Malopolska province in Poland. Overall this manuscript provides a strong resource dataset; however, there are some clarifications necessary to fully comprehend the data analysis.

Major concerns:

  1. Abstract: There is a disconnect between the results and conclusion text in that there are specific conclusions stated when the data are not described in the results

Author comments

Thank you for pointing this out. The reviewer is correct, and we have changed the abstract. However, we cannot add all information in result sections due to the abstract’s framework limitation (200 words).

  1. Table 3 needs a legend to explain what the p value is comparing. As presented, the p value appears to denote differences within a category (e.g. histological type). However, the results text seems to describe comparisons of one aspect of a category “In elderly patients, PA occurred less frequently, but WT was more common (p < 0.001)”. The p value is listed in the same row as PA, which makes this sentence very confusing. A similar issue exists for Tables 4 and 5.

Author comments

Thank you for reminding us about how important it is to present complex material in a concise and readily accessible way. However, all of the statistically significant differences were describe in the manuscript body. Also we try to keep the table layout uniform, for that reason the p-value in each column is listed in the same row.

  1. Table 4 is exactly the same as Table 3 and does not report the characteristics by sex

Author comments

Thank you for finding this mistake. We have changed the Table 4.

  1. Is there information on recurrence or 5 year follow-up?

Author comments

Thank you for your assessment. The paper does not contain an information on recurrence or 5-year follow up because it was not the aim of this study. It is also impossible to add each information in the article which has limited text framework.

  1. Discussion: obesity link with WT should include obesity rates in Malopolska province.

Author comments

As suggested by the reviewer, we have add information about obesity rate in Małopolska province from 2016 with proper reference (text line No 368-369).

Reviewer 3 Report

This a well-presented comprehensive study of the incidence of salivary gland tumours in southern Poland. The authors suggest that single-centre studies may not have significant statistical power but have herein presented such a study. The discussion of their findings would have been strengthened by discussing the limitations of their study and suggesting novel ideas for overcoming these limitations. 

An interesting finding from the study was the apparent change in the incidence of Warthin's tumours and Adenoid cystic carcinomas over the time period of the study. Whilst there was some discussion surrounding reasons for increase Warthin's incidence a similar discussion for adenoid cystic carcinomas was lacking.

There was also a discussion around surgical treatment for Warthin's without the real in-depth information that would allow the reader to understand why treatment might have change and the impact of such changes

Author Response

Reviewer 3

This a well-presented comprehensive study of the incidence of salivary gland tumours in southern Poland. The authors suggest that single-centre studies may not have significant statistical power but have herein presented such a study. The discussion of their findings would have been strengthened by discussing the limitations of their study and suggesting novel ideas for overcoming these limitations. 

An interesting finding from the study was the apparent change in the incidence of Warthin's tumours and Adenoid cystic carcinomas over the time period of the study. Whilst there was some discussion surrounding reasons for increase Warthin's incidence a similar discussion for adenoid cystic carcinomas was lacking.

There was also a discussion around surgical treatment for Warthin's without the real in-depth information that would allow the reader to understand why treatment might have change and the impact of such changes

Author comments

Thank you for your assessment. We have made the following changes in the discussion section (text line No 372-379 and 399-401). We have found also small decrease in adenoid cystic carcinoma incidence rate in the paper form Japan. However, it is hard to explain what is the reason of such a decrease. Probably it can be caused by better histological differential diagnosis nowadays. Also we have added some new information on higher incidence rate of Warthin’s tumor and possibility of “wait and see policy”.